# *De Novo* Transcriptome Assembly of Two *Microsorum* Fern Species Identifies Enzymes Required for Two Upstream Pathways of Phytoecdysteroids

**DOI:** 10.3390/ijms22042085

**Published:** 2021-02-19

**Authors:** Siriporn Sripinyowanich, Eui-Joon Kil, Sahanat Petchsri, Yeonhwa Jo, Hoseong Choi, Won Kyong Cho, Sukchan Lee

**Affiliations:** 1Department of Botany, Faculty of Liberal Arts and Science, Kasetsart University Kamphaeng Saen Campus, Nakhon Pathom 73140, Thailand; siriporn.srip@ku.th (S.S.); faassnps@ku.ac.th (S.P.); 2Department of Plant Medicals, Andong National University, Andong 36729, Korea; viruskil@anu.ac.kr; 3Research Institute of Agriculture and Life Sciences, College of Agriculture and Life Sciences, Seoul National University, Seoul 08826, Korea; yeonhwajo@gmail.com (Y.J.); bioplanths@snu.ac.kr (H.C.); 4Department of Integrative Biotechnology, Sungkyunkwan University, Suwon 16419, Korea

**Keywords:** fern, *Microsorum*, transcriptome, RNA-Seq, phytoecdysteroids

## Abstract

*Microsorum* species produce a high amount of phytoecdysteroids (PEs), which are widely used in traditional medicine in the Pacific islands. The PEs in two different *Microsorum* species, *M. punctatum* (MP) and *M. scolopendria* (MS), were examined using high-performance liquid chromatography (HPLC). In particular, MS produces a high amount of 20-hydroxyecdysone, which is the main active compound in PEs. To identify genes for PE biosynthesis, we generated reference transcriptomes from sterile frond tissues using the NovaSeq 6000 system. De novo transcriptome assembly after deleting contaminants resulted in 57,252 and 54,618 clean transcripts for MP and MS, respectively. The clean *Microsorum* transcripts for each species were annotated according to gene ontology terms, UniProt pathways, and the clusters of the orthologous group protein database using the MEGAN6 and Sma3s programs. In total, 1852 and 1980 transcription factors were identified for MP and MS, respectively. We obtained transcripts encoding for 38 and 32 enzymes for MP and MS, respectively, potentially involved in mevalonate and sterol biosynthetic pathways, which produce precursors for PE biosynthesis. Phylogenetic analyses revealed many redundant and unique enzymes between the two species. Overall, this study provides two *Microsorum* reference transcriptomes that might be useful for further studies regarding PE biosynthesis in *Microsorum* species.

## 1. Introduction

Species in the fern genus *Microsorum* in the family *Polypodiaceae* are major medicinal plants in Polynesian cultures [1]. In particular, the fronds and the rhizomes of *Microsorum* have been widely used as traditional remedies for pneumonia, gonorrhea, dislocations, and fractures and have antibacterial, analgesic, and anti-inflammatory properties [2,3,4,5].

*Microsorum* species produce a huge amount of phytoecdysteroids (PEs), which are bioactive compounds composed of 20-hydroxyecdysone, ecdysone, inokosterone, 2-deoxyecdysone, 2-deoxy-20-hydroxyecdysone, makisterone A, and makisterone C [6]. PEs control arthropod development and reproduction [7,8]. In addition, PEs resemble the brassinosteroids of plants in chemical structure, which are weak ecdysteroid antagonists in insects [9,10]. For example, plant-parasitic nematodes showed abnormal molting, immobility, reduced invasion, impaired development, and ultimately death after treatment with 20-hydroxyecdysone [11,12]. Furthermore, PE ingestion has been shown to promote growth in sheep and quail [13,14] and to stimulate carbohydrate metabolism and reduce rat hypoglycemia [15]. In addition, PEs are consumed as pseudosteroidal muscle enhancers by weightlifters and bodybuilders [5,16,17]. Currently, there are more than 140 dietary supplements containing PEs [6]. 

Previous studies demonstrated that the number of accumulated PEs was different among different *Microsorum* species [6,18,19]. It seems that the difference in the production of PEs in different *Microsorum* species might be caused by different genetic backgrounds. Although PEs are applied in diverse areas such as agriculture, food, and the pharmaceutical industry, knowledge of PE biosynthesis and regulatory mechanisms is limited due to the lack of genomic information in plants producing PEs [18]. 

Ferns are a type of vascular plant that produce spores in alternating gametophyte and sporophyte generations [20]. Although ferns are an important plant group, along with bryophytes, lycophytes, and gymnosperms, the available genomic information for ferns is limited because of their large genome, with an average size of 12 GB [21]. Recently, the genomes of two fern species, *Azolla filiculoides* and *Salvinia cucullata*, with relatively small genome sizes, were sequenced to elucidate land-plant evolution [22]. De novo transcriptome assembly based on RNA sequencing (RNA-Seq) allows us to generate reference transcriptomes for nonmodel plants, which facilitate the identification of new genes involved in various biological processes [23,24]. 

To date, many fern transcriptomes are currently available. For example, gene expression profiles between gametophyte and sporophyte phases in the *Polypodium amorphum* [25], transcriptomes for different developmental stages of sporangia in *Dryopteris fragrans*, and the root transcriptome of *Pteris vittata*, which is a hyperaccumulator of arsenic (As), have all been examined. In addition, many transcriptome studies have revealed the phylogeny and evolution of ferns [26,27,28].

In the present study, we examined PE compounds in two different *Microsorum* species based on high-performance liquid chromatography (HPLC). To identify genes for PE biosynthesis, we generated reference transcriptomes for the two *Microsorum* species using the NovaSeq 6000 system. Moreover, we identified several transcripts encoding for enzymes in MP and MS that were potentially involved in the mevalonate (MVA) and sterol biosynthetic pathways, which produce precursors for PE biosynthesis. In summary, we provided two *Microsorum* reference transcriptomes, which can be applied for the identification of genes or enzymes involved in PE biosynthesis. 

## 2. Results

### 2.1. Phenotypes of Two Microsorum Species

We used two different *Microsorum* species, *M. punctatum* (L.) Copel (2n = 72) and *M. scolopendria* (Burm. f.) Copel (2n = 72), which are abbreviated as MP and MS, respectively. MP has pale light green, leathery long fronds (leaf and stalk of fern), which are narrowly oblong to lanceolate (Figure 1a), while MS has deeply lobed leathery fronds with up to six lateral lobes that are light green in color (Figure 1b). In general, two different types of fronds can be found in *Microsorum* species. The sterile fronds of MP and MS do not bear sporangia, which are the cells in which spores are formed (Figure 1c,d), while the fertile fronds contain sporangia as shown in Figure 1e. In order to investigate the phylogeny of MP and MS, we generated a phylogenetic tree using chloroplast *rbcL* gene sequences (Figure 1f). MP and MS were in the same *Microsoroid* genera. 

### 2.2. Examination of PE Compounds from Two Microsorum Species

We examined compounds of PEs in two different *Microsorum* species based on HPLC analysis using a crude sample (2 mg dry weight) of *Microsorum* sterile fronds. The HPLC results of MP extract showed peaks with the retention times of 3.283, 6.167, and 9.167 min, which indicated three major PE compounds, namely ecdysone (0.32 mg/g), 20-hydroxyecdysone (0.17 mg/g), and makisterone (0.1 mg/g), respectively (Figure 2a). By contrast, MS extract had detected peaks with the retention times of 2.733, 3.267, 5.177, and 9.283 min, which were corresponding to 2-deoxy-20-hydroxyecdysone (0.18 mg/g), ecdysone (0.06 mg/g), 20-hydroxyecdysone (1.6 mg/g), and makisterone (0.09 mg/g), respectively (Figure 2b). Except for 2-deoxy-20-hydroxyecdysone, which was only identified in MS, the other three metabolites were commonly identified in both MP and MS (Figure 2c). However, the concentration of identified metabolites was different between MP and MS. For example, ecdysone was highly enriched in MP while MS contained a large amount of 20-hydrxyecdysone (Figure 2c). 

### 2.3. Library Preparation and RNA-Seq for the Generation of Two Microsorum Transcriptomes

The HPLC results demonstrated that the metabolites and the quantity of accumulated PEs in the two *Microsorum* species were different, indicating possible differences in the gene regulatory network associated with PE biosynthesis between the two species. We conducted transcriptome analyses for each species (Appendix A) to identify genes involved in PE biosynthesis in the two different *Microsorum* species. Frond samples from MP and MS were used for library preparation (Appendix A) and HPLC analysis (Appendix A). We generated a total of six libraries from the sterile frond tissues of MP and MS with three biological replicates, which were further paired-end sequenced using the NovaSeq 6000 system and resulted in 14.5 gigabytes of data (Appendix A). The number of sequenced reads ranged from 22 to 25 million (Table 1). The GC content ranged from 47.87% to 48.9%, and the Q20 percentage ranged from 98.05% to 98.24%.

### 2.4. De Novo Transcriptome Assembly and Taxonomy Classification

We conducted de novo transcriptome assembly for each *Microsorum* species using Trinity v2.4.0 [29] with a default k-mer value of 25. Data from the three libraries for each species were subjected to de novo transcriptome assembly generating a total of 131,351 and 106,153 transcripts for MP and MS, respectively (Table 2). To remove contaminated sequences, the two assembled transcriptomes were subjected to a BLASTX search against the National Center for Biotechnology Information (NCBI) nonredundant (NR) protein database using DIAMOND [30]. Based on the BLASTX results, we deleted transcripts assigned to microorganisms and animals. Finally, we obtained 57,252 and 54,618 clean transcripts for MP and MS, respectively (Table 2). The N50 and average contig length increased after removing contaminants. For example, the average contig length for MP increased from 846 bp in the preprocessed transcriptome to 1361 bp in the postprocessed transcriptome (Table 2). 

We evaluated the transcriptome completeness for two *Microsorum* species using BUSCO program (v2/v3) against the plant OrthoDB gene dataset. Of 1440 core plant genes, the number of core genes detected (Complete + Partial) was 1023 (71.04%) and 1037 (72.01%) for the postprocessing transcripts of MP and MS, respectively (Table 3). By contrast, the BUSCO scores were 95.7% and 94.9% for the preprocessed transcripts of MP and MS, respectively. We examined the size distribution of assembled transcripts (Figure 3a,b). Transcripts between 200 and 300 bp were dominant in both transcriptomes. The number of transcripts between 200 and 300 bp was much higher in MP (37,900 transcripts) than in MS (24,960 transcripts), and the number of transcripts that were more than 3 kb was higher in MS (6264 transcripts) compared to MP (4672 transcripts).

We analyzed the taxonomic classification for identified transcripts using the MEGAN6 program [31]. In total, 57,252 and 54,618 transcripts from MP and MS, respectively, were used for taxonomic classification. Of the 14 plant species identified, the top five plant species that accounted for more than 50% of the identified transcripts are shown in Figure 3c,d. A large number of transcripts were homologous to basal land plants including *Selaginella moellendorffii* (spikemoss), *Marchantia polymorpha* (liverwort), and *Physcomitrella patens* (moss). In MP, the transcripts assigned to *S. moellendorffii* (33%) were dominant followed by *Physcomitrella patens* (31%), *Marchantia polymorpha* (13%), and *Picea sitchensis* (12%) (Figure 3c). In MS, transcripts assigned to *S. moellendorffii* (29%) were dominant followed by *Marchantia polymorpha* (28%), *P. patens* (24%), and *P. sitchensis* (9%) (Figure 3d). We identified transcripts assigned to ferns including *Marsilea vestita* (waterclover)*, Dryopteris fragrans* (woodfern)*, Ceratopteris richardii* (triangle waterfern)*, Adiantum capillus-veneris* (maidenhair fern), *Pteris vittata* (ladder brake fern), and *Pteridium aquilinum* (bracken). There were only 64 transcripts (0.1%) with no blast hits.

### 2.5. Functional Annotation of Microsorum Transcripts

In order to annotate *Microsorum* transcripts, two *Microsorum* transcriptomes were subjected to a BLASTX search against the Uniref90 database the using Sma3s program with an *E*-value threshold of 10^−6^. Of 57,252 MP and 54,618 MS transcripts, 38,794 (67.96%) and 38,909 (71%), respectively, were annotated (https://doi.org/10.6084/m9.figshare.7856717 (accessed on 1 January 2021)). According to the molecular function, many transcripts in the two *Microsorum* species were assigned to gene ontology (GO) terms associated with catalytic activity, ion binding, and transporter activity (Figure 4). According to the cellular-component category, the intrinsic components of membrane and ribosome-associated were frequently identified. Among identified biological processes, transcripts assigned to metabolic process, transport, and RNA metabolic process were substantially enriched in both MP and MS. We compared two *Microsorum* transcriptomes using a TBLASTX search with E-value of 1e−5 as a cutoff. We found that 85% (46,152 transcripts) of total transcripts were commonly identified in both MP and MS. In total, 8466 transcripts were specific to MS. 

In addition, we identified enzymes assigned to 64 major metabolic pathways according to UniProt pathways. The three most abundant metabolic pathways were protein modification, lipid metabolism, and amino acid biosynthesis (Figure 5). In detail, the number of transcripts assigned to specific metabolic pathways varied between MP and MS. For example, the number of transcripts assigned to cofactor biosynthesis was 305 for MP and 327 for MS, whereas the number of transcripts assigned to carbohydrate degradation was 246 for MP and 294 for MS.

The annotation of *Microsorum* transcripts was performed based on clusters of orthologous groups in the COG protein database using MEGAN6 [31,32]. In total, 20,359 (35.56%) MP and 19,846 (36.34%) MS transcripts were classified into 25 COG categories with abbreviated terms, A to Z (Figure 6). The largest COG category was signal-transduction mechanisms (T) followed by post-translational modification, protein turnover, chaperones (O), and transcription (K). In contrast, only a few transcripts were assigned to COG categories associated with cell motility (N), extracellular and nuclear structures (W), and nuclear structure (Y). Interestingly, 937 (4.6%) and 798 (4%) transcripts for MP and MS, respectively, were assigned to secondary metabolite biosynthesis, transport and catabolism (Q) whereas 861 (4.23%) and 787 (3.94%) transcripts for MP and MS, respectively, were assigned to lipid transport and metabolism (I). In addition, we conducted differential gene-expression analysis by comparing MP to MS. The transcriptome similarity of two *Microsorum* species was very high (85%), and the MP contains a higher number of transcripts as compared to the MS. Therefore, we used the MP as a reference transcriptome for differential expression analysis. As a result, we identified 4516 up-regulated and 8081 down-regulated genes, respectively, by comparing MP to MS (Appendix A).

### 2.6. Identification of Microsorum Transcription Factors

Plant transcription factors (TFs) play important roles not only in gene expression but also in numerous biological processes. We identified TFs in *Microsorum* based on the BLASTP search against the PlantTFDB [33] as well as annotation results using Sma3s. We identified a total of 1852 and 1980 TFs for MP and MS, respectively, which were further assigned to 15 and 16 TF families (Figure 7). Among the TF families identified, ethylene response factor (ERF) was the dominant TF family followed by C2H2-type zinc finger (C2H2) and ligand-binding domain (LBD) families. In general, the number of identified TFs for each family between MP and MS was similar in general. However, the number of TFs for the GATA family was very high in MP (42 TFs) compared to MS (7 TFs), whereas the number of TFs for four TF families including WRKY, G2-like, NAC, and C3H families was substantially higher in MP than MS. According to Fisher’s exact test, five TF families such as GATA, TCP, G2-like, NAC, and C3H showed that the number of TFs was significantly different between MP and MS. By comparing MP to MS, we identified 54 up-regulated and 81 down-regulated TF genes, respectively (Appendix A).

### 2.7. Identification of Putative Microsorum Enzymes Involved in PE Biosynthesis

PEs are triterpenoid compounds derived from isopentenyl diphosphate (IPP) and dimethylallyl diphosphate (DMAPP), that are mainly synthesized via the mevalonate (MVA) pathway and sterol-biosynthetic pathway [18]. Based on previous studies, the PE biosynthetic pathway can be divided into three phases (Figure 8). The first phase consists of the MVA pathway generating the triterpenoid building units (IPP and DMAPP) (Figure 8a). The second phase is to generate lanosterol and cholesterol from squalene. The final phase is to produce ecdysone and 2-deoxy-20-hydroxyecdysone from lathosterol and cholesterol, respectively (Figure 8a). From the MP and MS transcriptomes, we identified putative enzymes involved in the PE-biosynthetic pathway (Table 4 and Appendix A). With regard to the MVA pathway, we identified 14 and 9 enzymes for MP and MS, respectively. In addition, 24 and 23 enzymes for MP and MS, respectively, were identified for the sterol biosynthetic pathway (Table 4). The number of identified enzymes in each step was different between MP and MS. For example, five MP and three MS enzymes encoding acetoacetyl-CoA transferase (AACT) were identified, whereas only a single enzyme encoding 3-hydroxy-3-methyglutaryl-CoA synthase (HMGS) was identified from both MP and MS.

### 2.8. Expression Analysis of Genes Involved in the MVA and Sterol-Biosynthetic Pathway

The production of PE compounds in two different *Microsorum* species might be dependent on the expression of genes involved in upstream PE biosynthesis. Therefore, we examined the expression of individual genes based on transcripts per million (TPM) values (Appendix A). Interestingly, several functionally redundant transcripts involved in PE biosynthesis were identified in each *Microsorum* transcriptome. For example, there were at least five transcripts encoding AACT protein, and their expression levels were varied (Table 4), and the TPM values of these five transcripts encoding AACT from MP ranged from 1.32 to 23.74, whereas the TPM values of the three transcripts encoding AACT from MS ranged from 3.06 to 75.44. Several transcripts encoding HMGS, HMGR, SQS, and EBP were highly expressed, while expression for the transcripts encoding MVK, PMK, SQE, CAS, ERG24, ERG26, DHCR24, and ERG3 were low. We compared the expression of individual transcripts required for MVA and sterol-biosynthetic pathways between the two species (Figure 8b and Table 4). Among functional redundant transcripts, we selected the transcript showing the highest expression. When compared to MP, seven transcripts encoding AACT1, HMGS1, IDI, FDS, DHCR24, EBP, and ERG3 in MS were up-regulated, while another 12 transcripts in MS were down-regulated (Figure 8b and Table 4). In order to confirm the RNA-Seq results, we carried out real-time RT-PCR for 11 transcripts encoding HMGS1, MVK, PMK, CAS, DHCR7, EBP, ERG-1, ERG-2, IDI, MVD, and SQS (Appendix A). All 11 real-time RT-PCR primer pairs in this study showed the clean melting curves indicating that real-time RT-PCR efficiency was in the range of 90–100%. Real-time RT-PCR showed that the expression of 10 transcripts, except the transcript encoding EBP, was up-regulated in MP compared to MS. The expression results for nine transcripts, except HMGS1 and IDI, were consistent between the real-time RT-PCR and RNA-Seq results. 

### 2.9. Phylogenetic Analyses of Enzymes Involved in MVA Pathway

In order to elucidate the evolutionary relationship between *Microsorum* and other plant species, we generated six phylogenetic trees for the enzymes involved in the MVA pathway. To generate phylogenetic trees, we collected available protein sequences for the MVA pathway from the protein database in NCBI. For AACT, there were at least six enzymes (MP_AACT1 to MP_AACT6) in MP and seven enzymes (MS_AACT1 to MS_AACT7) in MS (Figure 9a). Protein sequences for the AACT enzymes were highly conserved. Two HMGS enzymes in MP and a single HMGS enzyme in MS were grouped together in the same clade with those from *S. moellendorffii* and *Mesostigma viride* (Figure 9b). Four HMGR enzymes from MP and two HMGR enzymes from MS were in the same clade as other plant species, except for an HMGR enzyme from *Zea mays* (Figure 9c). With regard to mevalonate kinase (MVK), the phylogenetic tree showed that MP_MVK1 and MS_MVK1 were closely related to MVK from *S. moellendorffii*; however, MS_MVK2 from MS was very distantly related to other plant species (Figure 9d). In the phylogenetic tree of PMK, two PMK enzymes from MP with strong homology were closely related to other plant species in the same clade; however, two PMK enzymes from MS were very distantly related to other plant species (Figure 9e). There was only a single enzyme for MVD1 in both MP and MS, which was closely related to other plant species except for an MVD from *Triticum urartu* (Figure 9f). 

We identified 146 and 141 cytochrome P450 (CYPs) genes from two *Microsorum* transcriptomes using the BLASTX and HMMER programs (Appendix A). To reveal the phylogenetic relationship of the identified CYP proteins, we generated phylogenetic trees of CYP proteins from MP and MS, respectively, by including the 31 known *Drosophila* CYP proteins. The phylogenetic tree for CYP proteins of MP and MS showed several groups of CYP proteins (Appendix A). The most MP_CYP proteins were similar within the same clade, except for the MP_CYP25 protein (Appendix A). Interestingly, five MS_CYP proteins (MS_CYP63, MS_CYP70, MS_CYP34, MS_CYP67, and MS_CYP35) were in the same clade with other drosophila CYP proteins. Next, we generated a phylogenetic tree including all CYP proteins from these two *Microsorum* and *Drosophila* (Appendix A). As we expected, the *Microsorum* CYP proteins were phylogenetically different from the drosophila CYP proteins. The phylogenetic tree revealed that the number of identified CYP proteins in each subfamily was different between MP and MS.

## 3. Discussion

Ferns are different from other seed plants in various biological features. For example, they have large leaves (known as fronds) and a cluster of sporangia (known as sori), and they grow rhizomes instead of roots [34]. Although ferns are not of major economic importance and are typically used as ornamental plants, some fern species are beneficial for humans and have been used since ancient times [35]. In particular, secondary metabolites produced from some ferns have antioxidant, antibiotic, antitumor, and anti-inflammatory properties [36]. 

Currently, the genus *Microsorum* can be divided into at least 98 species [37]. At present, a limited number of studies associated with the members in the genus *Microsorum* has been reported. For example, some *Microsorum* species, including *M. fortunei* and *M. pteropus*, are cadmium (Cd) hyperaccumulators with Cd-accumulation capacities and Cd-resistance mechanisms [38,39]. Like other known ferns species, extracts of fronds and rhizomes in *Microsorum* species have been used as traditional medicines [4]. In addition, a previous study demonstrated that extracts of *M. grossum* could protect human skin against UV, suggesting its possible implications in cosmetic ingredients [2]. Furthermore, the phytochemical compounds of *Microsorum* extracts have been examined [5,40]. Similarly, we examined the phytochemical compounds of the two *Microsorum* species using HPLC, revealing that MP and MS were enriched with ecdysone and 20-hydroxyecdysone, respectively [41]. In particular, the amount of 20-hydroxyecdysone in MS was 9.4 times higher than in MP, which was consistent with an earlier study showing a high level of 20-hydroxyecdysone (0.2% of dry weight) in MS [5]. Our results suggested that there might be a strong difference in the regulation of PE biosynthesis among different *Microsorum* species. 

Although the compounds of PEs in *Microsorum* and their effects as medicines are well known, nothing is known about the key enzymes required for PE biosynthesis in *Microsorum* species. As an initial step in identifying enzymes involved in PE biosynthesis, we generated reference transcriptomes for two *Microsorum* species. Due to the large chromosome number and genome size of *Microsorum* species (2n = 72), we examined transcriptomes instead of genomes. Our de novo transcriptome assembly resulted in a high number of transcripts for the two *Microsorum* species. However, after deleting transcripts assigned to other organisms such as bacteria, fungi, and viruses, the proportion of clean transcripts associated with the plants was only 43.5% and 51.4% for MP and MS, respectively. This result suggested that under natural conditions plant samples can also be used as good material to investigate microorganisms present in the target plant. For example, several previous studies reported that plant transcriptomes sometimes contain sequences of plant viruses [42], fungi [43,44], and guest insects [45]. It is likely that RNA sequencing using oligo-d(T) could amplify mRNAs with poly(A) tails for eukaryotic cells as well as the virus genome with poly(A) tails. 

In contrast to other fern-related transcriptomes, our study focused on the identification of the key enzymes involved in the upstream pathways of PE biosynthesis involving the MVA and sterol-biosynthetic pathways. One interesting result was that there were many functionally redundant enzymes in the two *Microsorum* species, which might have been caused by whole-genome duplication events in fern genomes, as suggested in a recent study [22]. It is also possible that those functionally redundant enzymes were transcript isoforms that were predicted by the Trinity assembler. Moreover, gene-expression analyses indicated that functional redundant enzymes were differentially expressed suggesting a difference in their functional roles as enzymes. For example, flavonol synthase (FLS) plays a central role in flavonoid metabolism. Multiple isoforms encoding FLS might have different substrate specifications for the production of the flavonols, quercetin, and kaempferol. Of the examined genes involved in PE biosynthesis, the expression of many genes in MP was higher than that of the genes in MS. This result might be related with high level of a specific metabolite, for example, high concentration of ecdysone in MP. Interestingly, the expression of several genes such as *AACT1*, *HMGS1*, and *FDS* involved in the MVA pathway was higher in MS compared to those in MP. This result suggested that the activities of the several enzymes associated with MVA pathway might be higher in MS compared to MP. Again, two genes encoding DHCR24 and EBP were up-regulated in MS compared to MP. DHCR24 and EBP are required to synthesize lathosterol, which is a precursor of ecdysone. It is also likely that a transcript with a high level of expression plays a more important role than a transcript with a low level of expression. For instance, the content of 20-hydroxyecdysone (20E), which is the main active compound of PEs, in MS was 9.4-fold higher than that in MP. However, the amount of ecdysone in MP was five times higher than that in MS. We carefully suppose that the high amount of ecdysone in MP might be due to its nonconversion to 20-hydroxyecdysone. Thus, it is important to determine the genes or enzymatic processes associated with 20-hydroxyecdysone in the near future. Thus, gene-expression regulation associated with PE synthesis might be more complex than we thought. 

In fact, we did not identify novel enzymes directly involved in PE biosynthesis including ecdysone, 20-hydroxyecdysone, and 2-deoxy-20-hydroxyecdysone, which are also synthesized in animals [46]. An earlier study revealed that the P450 enzyme mediates the hydroxylation of ecdysone to 20-hydroxyecdysone in *Drosophila* [47]. Based on sequence similarity using the BLASTX and HMMER programs, we identified many P450 proteins in the two *Microsorum* transcriptomes; however, we could not identify a key P450 enzyme involved in PE biosynthesis due to the high number of P450 transcripts in the two *Microsorum* species. Possibly, the metabolic pathways in ferns are different from those in animal species. In addition, we also found that the enzymes in the same metabolic pathway varied between the two *Microsorum* species, although the majority of enzymes were shared between species. For example, both species had the PMK and MVK enzymes. However, two PMK enzymes in MP shared a strong similarity to other higher plants, whereas two PMK enzymes in MS were substantially different from other PMK enzymes in plants. In addition, MP had a single MVK enzyme, whereas MS had two MVK enzymes. MVK1 in MS might have the same functions as those in other plants; however, MVK2 in MS was distantly related in phylogenetic analyses. This result suggested a difference in the function of MVK2 compared to the MVK enzymes in other plants. A difference in the enzymes of the MVA pathway between different organisms has been demonstrated previously [48]. Transcriptome assembly tends to include alternative splicing and different fragments of the same transcript. Alternative splicing, producing several transcripts isoforms from a single gene, leads to protein diversity. Thus, the difference in the number of specific enzymes might be caused by the difference of alternatively spliced transcript numbers between these two species. It could regulate the secondary metabolism and quality of plants [49]. In our study, there were multiple genes for the same enzymes. Without the reference genome, it was difficult to get full-length transcripts using RNA-seq. Thus, it is important to confirm their sequences using PCR and Sanger-sequencing in the near future.

As shown in the phylogenetic analyses and gen-expression analyses, not only the number of enzymes but also their gene expression might contribute to the production of PEs between the two *Microsorum* species. It would suggest that the up-regulation of genes encoding AACT1, HMGS1, IDI, ERG24, and DHCR24 in MS compared to MP might be related to PE production. Similarly, an early study showed the positive effect of mevalonic acid and cholesterol on the synthesis of PEs in the fern *Polypodium* [50]. 

The classification of taxonomy showed that only about 10% of annotated *Microsorum* transcripts were homologous to those of ferns, indicating limited genetic information for fern species. In addition, a majority of *Microsorum* transcripts was homologous to *S. moellendorffii* (*Lycopodiophyta*), *P. patens* (*Bryophyta*), *M. polymorpha* (*Marchantiophyta*), and *P. sitchensis* (*Pinophyta*) suggesting there is a close evolutionary relationship between fern *Microsorum* species. 

In this study, we identified a large number of TFs (at least 1852 TFs for MP and 1980 transcripts for MS), which were further assigned into 15 and 16 TF families, respectively. However, the number of TFs in *Microsorum* was much higher than that in *S. moellendorffii* (665 TFs); even though the number of TF families in *Microsorum* was much lower than for *S. moellendorffii* (54 families). Our results were consistent with a recent study that demonstrated the duplication of fern TF genes [51]. Interestingly, no TF in the C3H family was identified in MP whereas 46 TFs of the C3H family were identified in MS. This result indicated the possibility of a strong difference in TF genes or their gene expression between the two different *Microsorum* species. Of the TF families identified, three TF families, ERF, bHLH, and MYB, which are involved in Jasmonate (JA) signaling, were abundantly present in the *Microsorum* species. JA-responsive TFs play an important role in the regulation of plant secondary metabolism [49]. A recent transcriptome study showed that the TFs and enzymes required for secondary metabolism were differentially expressed upon methyl JA treatment [52]. In particular, a previous study demonstrated that methyl JA stimulated the production of 20-hydroxyecdysone in cell-suspension cultures of *Achyranthes bidentate* [53]. Therefore, it might be of interest to find TFs that regulate the expression of genes involved in PE production for *Microsorum* species. 

In summary, we provided reference transcriptomes for two *Microsorum* species with different capacities for PE production. Our comparative transcriptome analyses suggested that multiple genes were involved in PE production and provided essential genetic information for *Microsorum* species as well as for the evolutionary study of ferns. 

## 4. Materials and Methods

### 4.1. Plant Materials and Sample Collection

Two *Microsorum* species, *M. punctatum* (Linn.) Copel and *M. scolopendria* (Burm.f.) Copel, were grown in compost soil without fertilizer in the experimental plot at Kasetsart University Kamphaeng Saen Campus, Nakhon Pathom, Thailand. Two *Microsorum* species could be established in the same core Microsoroid. Mature sporophyte frond tissues of two-months-old *Microsorum* species were sampled independently. One sterile frond (a leaf lacking sporangia) was rinsed with distilled water and collected as a single biological replicate for RNA extraction. Frozen samples with liquid nitrogen were stored at −80 °C. 

### 4.2. Measurement of Phytoecdysteroids

The extraction of phytoecdysteroids was performed in accordance with a methodology described previously [5]. In brief, approximately 50 mg sterile fronds were dried in an oven at 55 °C for 16 h and grinded by mortar and pestle, 2 mg samples were extracted in 95% ethanol in a soxhlet apparatus for 6 h. The ethanol extracts were evaporated by rotary evaporator at 60 °C. The residue was dissolved in methanol and vortexed with hexane. The methanol extracts were evaporated at 60 °C in a hot-air oven. The residue was dissolved in HPLC-grade methanol (1 mL). The supernatant was filtered through a 0.45 μM nylon membrane filter (Millipore, Darmstadt, Germany) and then dried at room temperature in laminar hood. The residues were re-dissolved in HPLC-grade methanol and analyzed by an HPLC system (Water Corporation, Miford, MA, USA) equipped with an ultraviolet (UV) detector set at wavelength of 245 nm. The separation was achieved on a Waters Spherisorb C18 ODS2 column (4.6 mm × 250 mm i.d., 5 μm) with Spherisorb S5 ODS2 guard column (4.6 mm × 10 mm C18 ODS2 i.d., 5 μm) at 40 °C. The mobile phase consisted of either a mixture of HPLC grade 14% acetonitrile in 2% acetic acid at a flowrate of 1 mL/min. The volume of the injected samples was 20 μL, running for 20 min. The identification of each compound was based on a combination of retention time and spectral matching. Calibration of the system with known amounts of these compounds was performed using standard samples (Sigma-Aldrich, St. Louis, MO, USA). The stock solutions of standard were prepared and applied in triplicate onto the HPLC. The peaks areas were recorded, and the calibration curve were prepared by plotting peak areas against concentrations.

### 4.3. Total RNA Extraction and Transcriptome Sequencing

Total RNA was extracted using NucleoSpin RNA (Macherey-Nagel, Düren, Germany) according to the manufacturer’s instructions. The quality and quantity of extracted RNA were measured by gel electrophoresis, ND-1000 Nanodrop spectrophotometer (Thermo Scientific, Waltham, MA, USA), and Agilent 2100 Bioanalyzer (Aligent, Santa Clara, CA, USA). At least 20 μg of total RNA at a concentration of ≥ 400 ng/μL, OD260/280 = 1.8–2.2, RNA 28S: 18S ≥ 1.0, and RNA Integrity Number (RIN) ≥ 7.0 was used for library preparation. The high-quality total RNAs were used for library preparation using a TruSeq RNA Library Prep Kit v2 (Illumina, San Diego, USA) following the manufacturer’s instructions. A total of six libraries from three biological replications representing two *Microsorum* species were paired-end sequenced by the Illumina NovaSeq 6000 system at Macrogen in Seoul, Republic of Korea.

### 4.4. De Novo Transcriptome Assembly and Deletion of Contaminated Transcripts

All six fastq files derived from three different libraries for each species were subjected to de novo transcriptome assembly using a Trinity pipeline and the Trinity program (version 2.0.2, released 22nd January 2015) with default parameters (kmer size 25) (http://trinityrnaseq.github.io/ (accessed on 1 January 2021)) [29]. We used all transcript isoforms predicted by the Trinity assembler in our study. The transcriptome completeness for two *Microsorum* species was assessed using BUSCO (v2/v3) program implemented in gVolante (https://gvolante.riken.jp/ (accessed on 1 January 2021)) against the plant OrthoDB gene dataset. Assembled transcripts were used for the BLASTX search against the NR protein dataset of the NCBI using the DIAMOND program (https://ab.inf.uni-tuebingen.de/software/diamond (accessed on 1 January 2021)) [30] with default parameters (E-value 1e−5 as a cutoff). The output file in dmnd file format from DIAMOND was subjected to the MEGAN6 program, which is used for taxonomic analysis based on NCBI taxonomy (https://ab.inf.uni-tuebingen.de/software/megan6 (accessed on 1 January 2021)) [31]. Based on the taxonomy results by MEGAN6, we deleted contaminated transcripts such as sequences from animals, bacteria, fungi, and viruses. As a result, only plant assigned transcripts and unassigned transcripts were included in the annotation of the *Microsorum* transcriptome. 

### 4.5. Annotation of Microsorum Transcripts and Prediction of Open Reading Frames (ORFs)

Clean *Microsorum* transcripts from the two different species were used for gene annotation. For annotation, we used two different methods. The first was a combination of BLASTX with E-value of 1e−5 as a cutoff against the NR database using DIAMOND followed by functional annotation using InterPro2GO, SEED, and eggNOG programs implemented in MEGAN6. The second was gene annotation using Sma3s using the following command (./sma3s.pl -i query_dataset.fasta -d uniref90.fasta -nucl –goslim) (https://github.com/UPOBioinfo/sma3s (accessed on 1 January 2021)) [54]. The Sma3s conducts a BLASTX against the UniProt database (uniref90.fasta) using the normal NCBI BLAST+ program. Based on the BLASTX result, transcripts were annotated according to GO terms (http://www.geneontology.org/ (accessed on 1 January 2021)), Swiss-Prot keywords, and metabolic pathways by Sma3s. Moreover, we further predicted coding regions within the assembled transcripts by the TransDecoder program using the following command (TransDecoder.LongOrfs -t target_transcripts.fasta) embedded in the Trinity software distribution. The longest transcripts in each isoform were used for the prediction of open reading frames (ORFs).

### 4.6. Gene-Expression Analysis of the Selected Microsorum Genes

To evaluate the expression level of the selected *Microsorum* genes, we calculated transcripts per million (TPM) values. For that, we combined fastq files from three different libraries according to *Microsorum* species. The combined raw sequence reads were mapped on the assembled transcriptome for MP and MS, respectively, using Burrows–Wheeler aligner (BWA) (http://bio-bwa.sourceforge.net/ (accessed on 1 January 2021)) program using the following command: (bwa mem ref.fa reads.fq > aln-se.sam) with default parameters. The two mapped sam files were subjected to eXpress program (https://pachterlab.github.io/eXpress/ (accessed on 1 January 2021)) with the following command (express target_seqs.fasta aligned_reads.sam) to calculate TPM values. The calculated transcript abundance for identified transcripts in MP and MS can be found in Appendix A.

For differentially expressed gene (DEG) analysis, all six raw data were aligned on the transcriptome for MP by BWA aligner. The aligned read numbers for each transcript were calculated by eXpress program. The numbers of aligned reads were subjected to DEG analysis using DESeq2 program (https://bioconductor.org/packages/release/bioc/html/DESeq2.html (accessed on 1 January 2021)). DEGs were identified based on fold changes more than two times and *p*-values less than 0.01 by comparing MP to MS. The calculated fold changes and adjusted *p*-values can be found in Appendix A. 

### 4.7. Real-Time RT-PCR Analysis

In order to validate the results of RNA-Seq, we conducted real-time RT-PCR for the 11 selected *Microsorum* genes encoding HMGS1, MVK, PMK, MVD, IDI, SQS, CAS, ERG-1, ERG-2, EBP, and DHCR7. The experiment was performed on the same set of RNA samples used for RNA-Seq. We designed primers for all the transcripts that were annotated for the same gene. Quantitative reverse transcription polymerase chain reaction (qRT-PCR) was employed using iQ SYBR Green Supermix (Bio-Rad, Hercules, CA, USA) and CFX96 (Bio-Rad). Data were analyzed using Rotor-Gene Q series software version 2.3.1 (Qiagen). To identify internal control genes of *Microsorum*, we carried out a BLASTN search using four *Arabidopsis* internal gene sequences (*ACT11*, *ACT2*, *TUA*, and *UBQL*) against MP and MS transcriptomes [55]. If there is more than one transcript for an enzyme, we selected the transcript with the highest TPM value for primer design. We designed primers and conducted real-time RT-PCR. Finally, primers amplifying a single amplicon were selected as the internal control gene sequence. The real-time RT-PCR results were normalized with *ACT11* and *UBQL* genes. Real-time RT-PCR was conducted as follows: the first step at 95 °C for 3 min, followed by 40 cycles of denaturation at 95 °C for 10 s, annealing at 55 °C for 30 s, and elongation at 72 °C for 30 s. Fluorescence measurements were conducted once per cycle at the end of the annealing temperature. Primer sequences for real-time RT-PCR are shown in Appendix A. The reactions were conducted in technical and biological triplicate.

### 4.8. Identification of Transcription Factors and P450 Genes

In order to identify TF in *Microsorum*, we carried out BLASTX search with E-value of 1e−5 as a cutoff against the database containing known TF proteins derived from PlantTFDB [33]. Based on BLASTX results and annotation results by Sma3s, we classified identified TFs according to TF families. 

To find cytochrome P450 (CYPs) genes from two *Microsorum* transcriptomes, the obtained reference transcriptomes were used for BLASTX search with E-value of 1e−5 as a cutoff against P450 proteins of *Drosophila melanogaster* (Appendix A). After that, only sequences containing p450 domain (PF00067) were identified based on HMMER (version 3.3.1) search (http://hmmer.org/ (accessed on 1 January 2021)). The identified CYP proteins from MP and MS were listed in Appendix A.

### 4.9. Phylogenetic Analysis

For phylogenetic analysis, we selected six proteins, AACT, HMGS, HMGR, MVK, PMK, and MVD, which are involved in the MVA pathway. First, we conducted a search using BLASTP and the six reference protein sequences against the two *Microsorum* reference proteomes, respectively. The obtained protein sequences were aligned by ClustalW implemented in the MEGA7 program [56]. The aligned sequences were subjected to phylogenetic tree construction with the maximum likelihood (ML) method and 1000 bootstrap values using MEGA7. 

For the construction of phylogenetic trees of CYP proteins, the identified *Microsorum* CYP proteins and drosophila CYP proteins were aligned together using MAFFT program with G-INS-1 option (https://mafft.cbrc.jp/ (accessed on 1 January 2021)). The aligned protein sequences were trimmed by Trimal program with automated1 option (http://trimal.cgenomics.org/ (accessed on 1 January 2021)). The trimmed protein sequences were subjected to IQ-Tree program for the phylogenetic tree construction with maximum-likelihood method (http://www.iqtree.org/ (accessed on 1 January 2021)). The LG + I + G4 substitution model was used for all CYP phylogenetic tree construction. The generated phylogenetic trees were visualized by Figtree version 1.4.4 (https://github.com/rambaut/figtree/releases (accessed on 1 January 2021)).

## Figures and Tables

**Figure 1 ijms-22-02085-f001:**
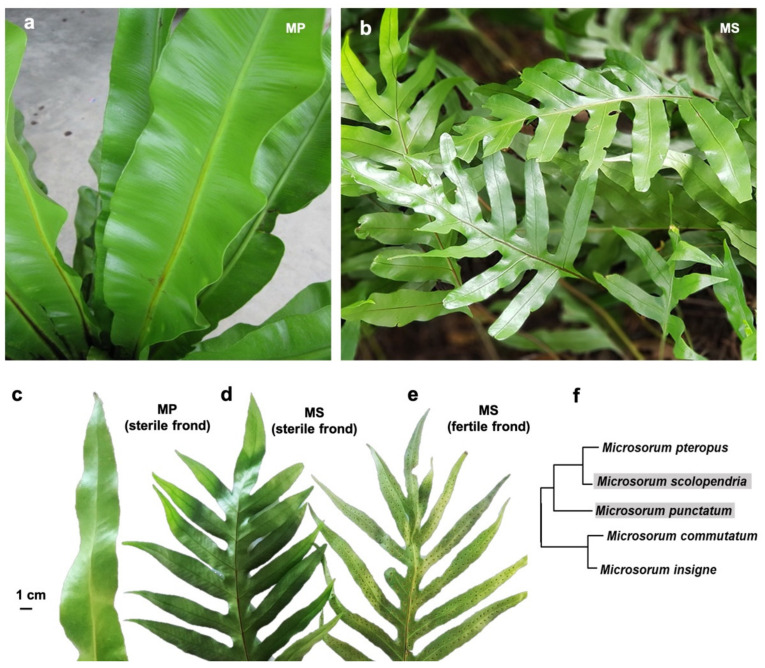
Frond images of *M. punctatum* (MP) (**a**) and *M. scolopendria* (MS) (**b**). A largely developed sterile frond of MP (**c**). Two types of fully developed fronds, sterile (**d**) and fertile fronds (**e**) for MS. Scale bar indicates 1 cm. (**f**) Phylogenetic tree using *rbcL* gene sequences showing the phylogenetic relationship of the two *Microsorum* species in this study with three other *Microsorum* species.

**Figure 2 ijms-22-02085-f002:**
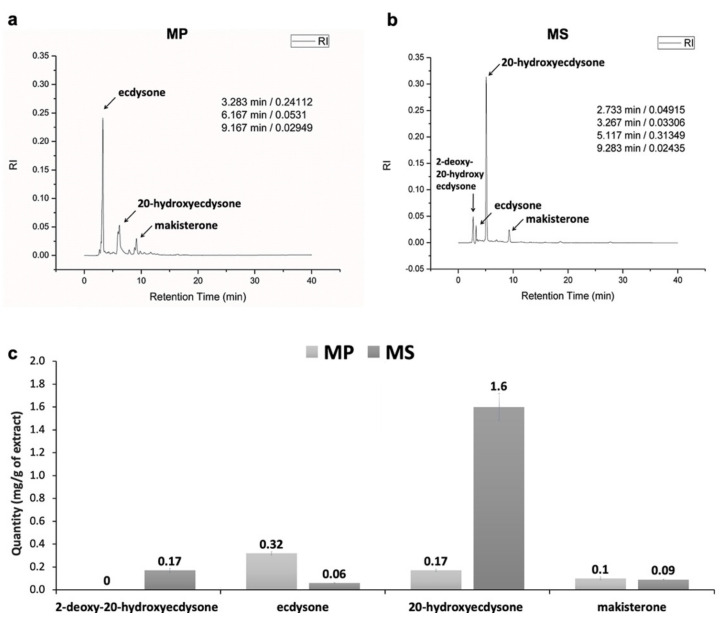
High-performance liquid chromatography (HPLC) analysis of phytoecdysteroids (PEs) extracted from sterile frond tissues of two different *Microsorum* species, *M. punctatum* (MP), and *M. scolopendria* (MS). Most significant PEs for MP (**a**) and MS (**b**) were indicated by the name of the respective metabolite. Quantification of phytoecdysteroids compounds (**c**) in sterile fronds of MP and MS. Error bars represent mean values with standard errors (*n* = 3).

**Figure 3 ijms-22-02085-f003:**
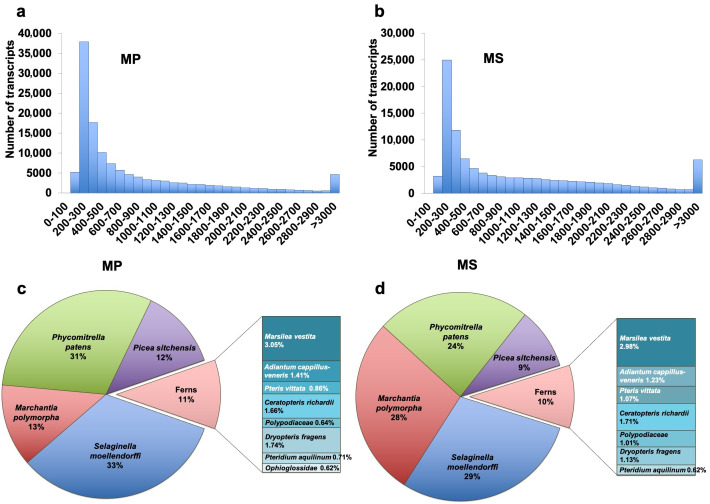
Bar graph visualizing the sequence length distribution of identified transcripts for *M. punctatum* (MP) (**a**) and *M. scolopendria* (MS) (**b**). Assembled transcripts of more than 200 bp were subjected to BLASTX search against the nonredundant (NR) database in National Center for Biotechnology Information (NCBI) using DIAMOND. Of the transcripts assigned to plant species, the major plant species were visualized with the respective proportion of MP (**c**) and MS (**d**). Transcripts assigned to ferns were further classified in detail.

**Figure 4 ijms-22-02085-f004:**
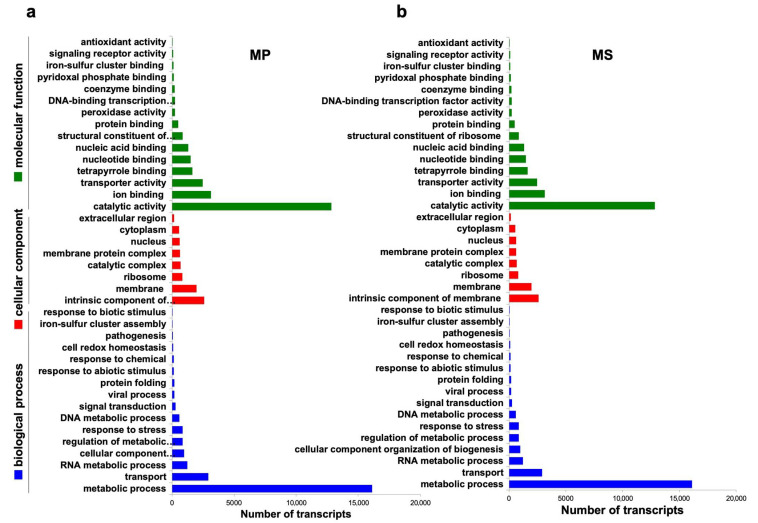
Gene ontology distribution of the identified *Microsorum* transcripts according to biological process (blue), cellular component (red), and molecular function (green) for *M. punctatum* (MP) (**a**) and *M. scolopendria* (MS) (**b**).

**Figure 5 ijms-22-02085-f005:**
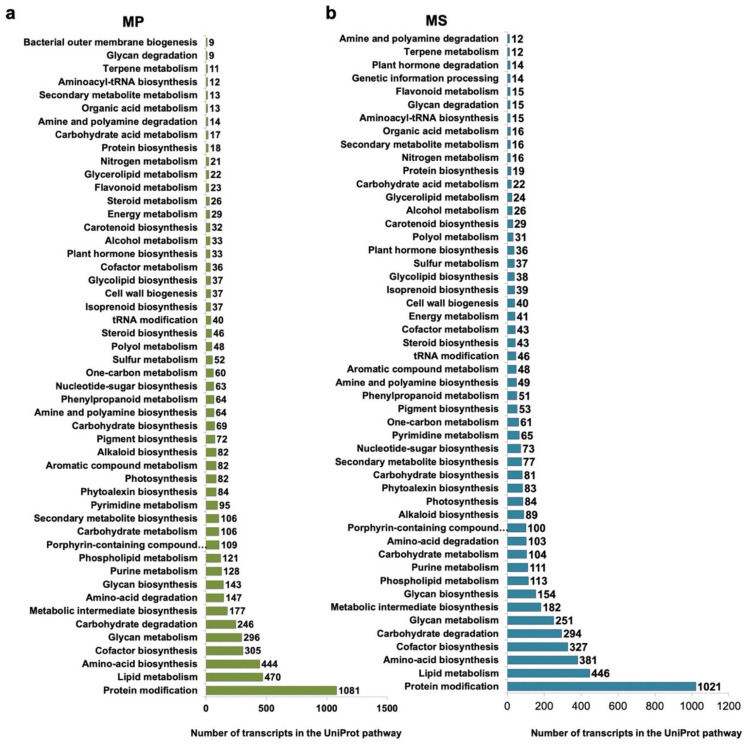
Functional annotation of *Microsorum* transcripts according to UniProt pathways. Top 50 UniProt pathways for *M. punctatum* (MP) (**a**) and *M. scolopendria* (MS) (**b**) based on the number of assigned transcripts. Each transcript was annotated according to the UniProt pathway using the Sma3s program.

**Figure 6 ijms-22-02085-f006:**
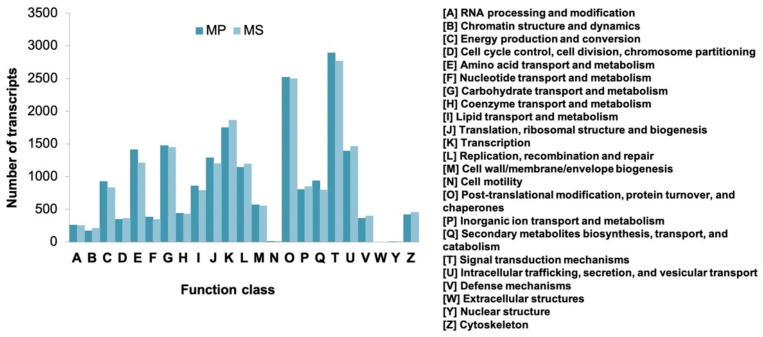
Clusters of orthologous group classification of *Microsorum* transcripts. *Microsorum* transcripts were assigned to clusters of orthologous groups for *M. punctatum* (MP) and *M. scolopendria* (MS) using MEGAN6.

**Figure 7 ijms-22-02085-f007:**
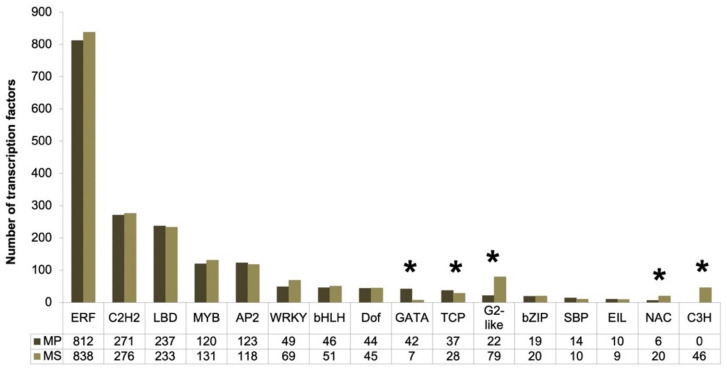
Bar graph showing the number of transcription factors according to individual transcription factor family for *M. punctatum* (MP) and *M. scolopendria* (MS). *Microsorum* transcription factors were identified using the functional annotation of the Sma3s program and BLASTX search against a plant transcription factor database. * indicates statistical significance less than 0.001 according to Fisher’s exact test at *p* < 0.05.

**Figure 8 ijms-22-02085-f008:**
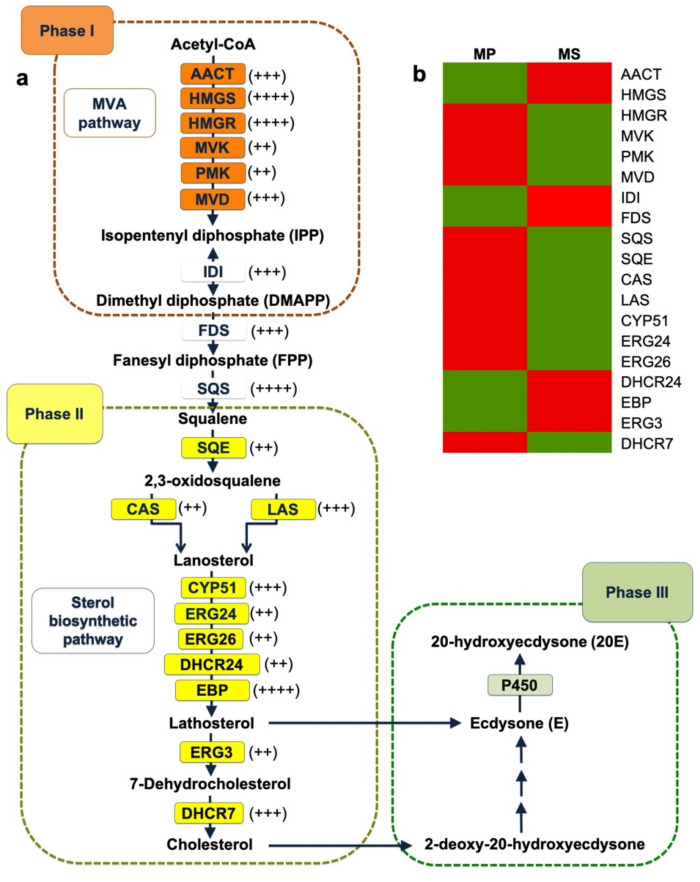
Putative enzymes involved in the phytoecdysteroids (PEs) biosynthetic pathway. (**a**) Proposed PEs biosynthetic pathway in *Microsorum* species. Orange, yellow, and green boxes represented proteins involved in the MVA, sterol biosynthesis, and 20E biosynthetic pathways, respectively. Expression level of individual genes was indicated by + sign; < 5 (+), 5–20 (++), 20–50 (+++), >50 (++++) based on average transcripts per million (TPM) values, and the highest TPM value for each enzyme was used. (**b**) The heatmap of the TPM distribution of all transcripts involved in the PEs biosynthesis was visualized by red and green, indicating up-regulated and down-regulated genes, respectively. Detailed information for enzymes identified is provided in Table 4.

**Figure 9 ijms-22-02085-f009:**
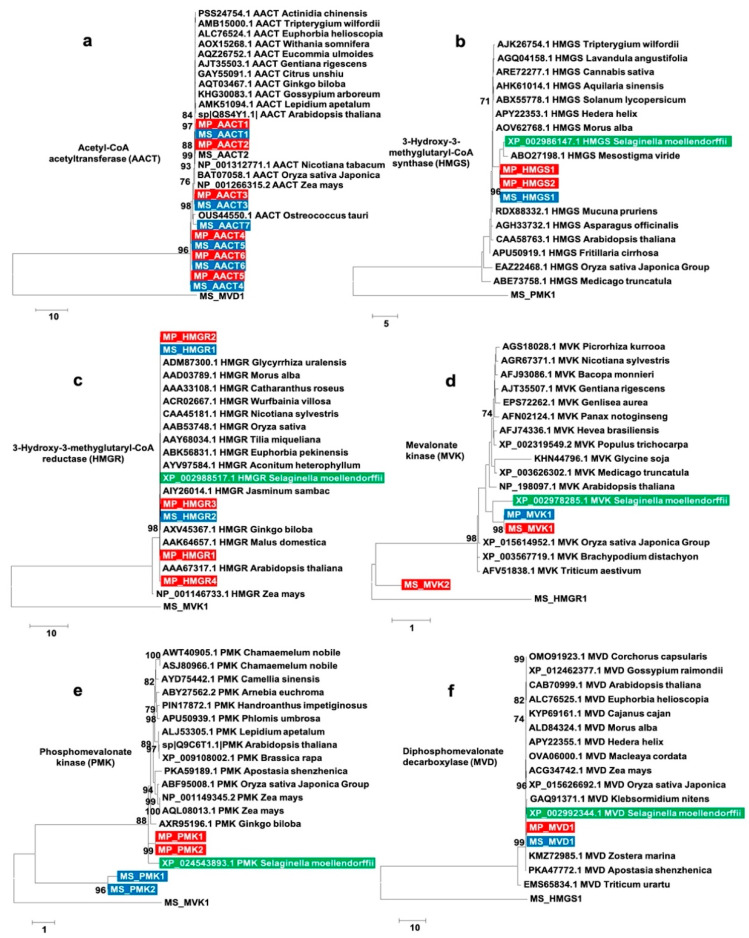
Phylogenetic trees of six putative enzymes involved in mevalonate (MVA) pathway; AACT (**a**), HMGS (**b**), HMGR (**c**), MVK (**d**), PMK (**e**), and MVD (**f**). Their protein sequences were identified from two *Microsorum* species and other plant species from NCBI protein database. Using available protein sequences for the six enzymes, BLASTP was conducted against two *Microsorum* proteomes. We also selected highly homologous proteins from the protein database for individual enzymes based on BLASTP results. The obtained protein sequences were aligned using ClustalW. The phylogenetic trees were constructed using the maximum-likelihood (ML) method with MEGA ver.7.0 software and 1000 bootstraps. The *Microsorum* protein sequences for individual enzymes are provided in Appendix A. Red and blue colors represent proteins identified from MP and MS, respectively. Proteins from *Selaginella moellendorffii* are represented in green. MS_MVD1, MS_PMK1, MS_MVK1, MS_HMGR1, MS_MVK1, and MS_HMGS1 were used as an outgroup for the phylogenetic trees of AACT, HMGS, HMGR, MVK, PMK, and MVD, respectively.

**Table 1 ijms-22-02085-t001:** Summary of paired-end sequencing for two *Microsorum* species.

Name of Library	Size of Raw Data (bp)	Total Number of Reads	GC Content (%)	Q20 (%)	Accession Number in SRA Database
MP_R1	2,423,192,808	23,992,008	48.90	98.05	SRR8480531
MP_R2	2,275,984,702	22,534,502	47.87	98.17	SRR8480553
MP_R3	2,567,004,688	25,415,888	47.94	98.22	SRR8480612
MS_R1	2,547,634,100	25,224,100	48.43	98.18	SRR8480631
MS_R2	2,340,604,906	23,174,306	48.36	98.22	SRR8480739
MS_R3	2,432,464,810	24,083,810	48.18	98.24	SRR8480744

A total of six libraries were prepared for *M. punctatum* (MP) and *M. scolopendria* (MS). For each species, three different libraries were paired-end sequenced by the NovaSeq 6000 system. Raw sequence data were deposited in the SRA database of NCBI with respective accession numbers.

**Table 2 ijms-22-02085-t002:** Summary of de novo assembled transcriptomes for two *Microsorum* species.

Read Statistics	*M. punctatum*		*M. scolopendria*	
	Preprocessing	Postprocessing	Preprocessing	Postprocessing
Total assembled bases	111,129,164	77,952,463	115,715,163	90,302,902
Number of transcripts	131,351	57,252	106,153	54,618
N50	1493	1934	1882	2148
Median transcript length	453	1099	660	1459
Average transcript length	846	1361	1090	1653
Total number of genes	80,026	27,243	64,098	23,175
GC content (%)	45.64	45.79	45.52	45.71

Preprocessing indicates de novo assembled transcriptome by Trinity without any filtering whereas post-processing indicates *Microsorum* transcriptome after contaminants are deleted. The number of transcripts indicates the number of whole transcripts while the number of genes indicates the number of representative genes.

**Table 3 ijms-22-02085-t003:** Summary of transcriptome completeness for two *Microsorum* species using BUSCO (v2/v3) program against the plant OrthoDB gene dataset.

	*M. punctatum*	*M. scolopendria*
Total number of core genes queried	1440	1440
Number of core genes detected (complete)	952 (66.11%)	986 (68.47%)
Number of core genes detected (complete + partial)	1023 (71.04%)	1037 (72.01%)
Number of missing core genes	417 (28.96%)	403 (27.99%)
Average number of orthologs per core genes	1.77	1.99
% of detected core genes that have more than 1 ortholog	52.73	68.97

**Table 4 ijms-22-02085-t004:** List of identified *Microsorum* enzymes involved in the mevalonate (MVA) pathway and sterol biosynthetic pathway, which might be responsible for PE biosynthesis.

Pathway	Enzymes	Enzyme no.	No. of Proteins		MP			MS		
			MP	MS	Unigene ID	Size	TPM	Unigene ID	Size	TPM
MVA	Acetoacetyl-CoA transferase (AACT)	2.3.1.9	5	3	MP_TRINITY_ DN32725_c0_g1_i1	1254	8.97	MS_TRINITY_ DN30309_c0_g1_i1	2001	13.79
					MP_TRINITY_ DN33165_c0_g2_i2	739	1.32	MS_TRINITY_ DN32401_c0_g2_i1	1792	75.44
					MP_TRINITY_ DN34009_c1_g1_i1	1774	1.53	MS_TRINITY_ DN36240_c0_g1_i2	1523	3.06
					MP_TRINITY_ DN38258_c0_g1_i2	755	23.74			
					MP_TRINITY_ DN42159_c0_g1_i1	1851	21.94			
	3-Hydroxy-3-methyglutaryl-CoA synthase (HMGS)	2.3.3.10	1	1	MP_TRINITY_ DN41084_c1_g4_i2	1751	22.11	MS_TRINITY_ DN32809_c0_g1_i1	1969	108.60

	3-Hydroxy-3-methyglutaryl-CoA reductase (HMGR)	1.1.1.34	4	2	MP_TRINITY_ DN38567_c1_g2_i1	2322	6.24	MS_TRINITY_ DN12064_c0_g2_i1	2498	9.16
					MP_TRINITY_ DN40805_c0_g1_i1	2429	7.49	MS_TRINITY_ DN35244_c0_g1_i1	2708	84.46
					MP_TRINITY_ DN43604_c0_g1_i1	636	4.87			
					MP_TRINITY_ DN44231_c0_g4_i1	2196	245.25			
	Mevalonate kinase (MVK)	2.7.1.36	1	1	MP_TRINITY_ DN35569_c0_g1_i1	1503	28.41	MS_TRINITY_ DN31620_c0_g1_i1	1622	10.26
	Phosphomevalonate kinase (PMK)	2.7.4.2	1	1	MP_TRINITY_ DN40957_c0_g1_i3	2108	13.19	MS_TRINITY_ DN31249_c0_g1_i1	2212	10.26
	Diphospho mevalonate decarboxylase (MVD)	4.1.1.33	2	1	MP_TRINITY_ DN18033_c0_g1_i1	471	1.09	MS_TRINITY_ DN32042_c0_g1_i1	1604	29.62
					MP_TRINITY_ DN38695_c0_g1_i1	1710	37.22			
Sterol biosynthesis	isopentenyl diphosphate isomerase (IDI)	5.3.3.2	1	1	MP_TRINITY_ DN40402_c1_g1_i3	1297	21.93	MS_TRINITY_ DN30481_c0_g1_i3	1244	48.19
	farnesyl diphosphate synthase (FDS)	2.5.1.10	1	2	MP_TRINITY_ DN38904_c0_g1_i1	1662	40.55	MS_TRINITY_ DN30957_c0_g1_i2	1382	7.46
								MS_TRINITY_ DN31472_c0_g1_i3	242	63.08
	squalene synthase (SQS)	2.5.1.21	1	2	MP_TRINITY_ DN38844_c0_g1_i1	1521	119.78	MS_TRINITY_ DN31427_c0_g1_i1	1456	46.17
								MS_TRINITY_ DN45369_c0_g1_i1	558	1.09
Sterol biosyn-thesis	squalene monooxygenase (SQE)	1.14.13.132	2	2	MP_TRINITY_ DN39062_c0_g1_i1	2019	18.31	MS_TRINITY_ DN31999_c0_g1_i2	1056	3.06
					MP_TRINITY_ DN43491_c0_g1_i3	2273	6.95	MS_TRINITY_ DN33834_c0_g1_i2	1926	4.83
	lanosterol synthase (LAS)	5.4.99.7	3	2	MP_TRINITY_ DN23554_c0_g1_i1	680	1.44	MS_TRINITY_ DN248_c0_g1_i1	771	3.14
					MP_TRINITY_ DN32916_c0_g1_i1	560	46.28	MS_TRINITY_ DN31641_c0_g1_i1	778	32.32
					MP_TRINITY_ DN37953_c0_g1_i2	2966	1.35			
	cycloartenol synthase (CAS)	5.4.99.8	1	1	MP_TRINITY_ DN37919_c0_g1_i1	2937	24.17	MS_TRINITY_ DN35135_c0_g1_i1	2970	12.96
	sterol 14- demethylase (CYP51)	1.14.13.70	3	3	MP_TRINITY_ DN38602_c0_g2_i1	1998	36.73	MS_TRINITY_ DN31074_c0_g2_i1	2047	2.19
					MP_TRINITY_ DN39045_c0_g2_i1	1555	2.98	MS_TRINITY_ DN33684_c0_g1_i1	1835	22.62
					MP_TRINITY_ DN39045_c0_g2_i3	529	1.80	MS_TRINITY_ DN45802_c0_g1_i1	266	1.09
	δ14-sterol reductase (ERG24)	1.3.1.70	1	1	MP_TRINITY_ DN37114_c0_g1_i1	1628	6.47	MS_TRINITY_ DN36333_c2_g1_i1	1591	4.45
	sterol-4α- carboxylate 3-dehydrogenase (ERG26)	1.1.1.170	2	2	MP_TRINITY_ DN42534_c0_g2_i1	1857	26.34	MS_TRINITY_ DN34567_c0_g1_i1	1325	10.97
					MP_TRINITY_ DN44751_c0_g2_i1	1540	16.40	MS_TRINITY_ DN35016_c1_g1_i1	2366	3.25
	δ24-sterol reductase (DHCR24)	1.3.1.72	5	2	MP_TRINITY_ DN31959_c0_g1_i1	2111	4.07	MS_TRINITY_ DN28185_c0_g1_i1	1932	3.69
					MP_TRINITY_ DN32815_c0_g1_i2	733	2.94	MS_TRINITY_ DN34498_c0_g1_i1	2309	13.39
					MP_TRINITY_ DN48554_c0_g1_i1	210	3.11			
					MP_TRINITY_ DN51472_c0_g1_i1	258	0.45			
					MP_TRINITY_ DN58603_c0_g1_i1	220	2.04			
	cholestenol δ-isomerase (EBP)	5.3.3.5	1	2	MP_TRINITY_ DN33408_c1_g1_i1	973	81.36	MS_TRINITY_ DN24433_c0_g1_i1	868	5.80
								MS_TRINITY_ DN27664_c0_g1_i1	941	113.34
	δ7-sterol 5- desaturase (ERG3)	1.14.19.20	1	1	MP_TRINITY_ DN34658_c0_g1_i1	791	7.50	MS_TRINITY_ DN26674_c0_g1_i1	1330	9.39
	7-dehydro cholesterol reductase (DHCR7)	1.3.1.21	2	2	MP_TRINITY_ DN37028_c0_g1_i1	937	22.02	MS_TRINITY_ DN27356_c0_g1_i1	817	9.46
					MP_TRINITY_ DN39419_c1_g1_i2	2247	53.09	MS_TRINITY_ DN31977_c0_g1_i1	2023	30.49

The name of the enzyme (Abbreviation), enzyme commission (EC) number, number of enzymes in *M. punctatum* (MP) and *M. scolopendria* (MS), transcript ID, and TPM value are listed. TPM value for each transcript was the average value of three replicates.

## Data Availability

The raw dataset in this study will be available, upon publication, in the Sequence Read Archive (SRA) repository with accession numbers SRR8480531, SRR8480553, SRR8480612, SRR8480631, SRR8480739, and SRR8480744. The assembled clean transcripts, gene annotation, and predicted protein sequences are freely available in figshare (https://doi.org/10.6084/m9.figshare.7856717).

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
