# Peer review of "De Novo Transcriptome Assembly of Two Microsorum Fern Species Identifies Enzymes Required for Two Upstream Pathways of Phytoecdysteroids"

_ijms, 2021, doi:10.3390/ijms22042085_

Round 1
Reviewer 1 Report
In this manuscript, the authors employed HPLC and transcriptome analyses to identify genes that are required for phytoecdysteroids (PE) biosynthesis in two Microsorum fern species.
Overall, I think this manuscript reads well. The authors de novo assembled 57,252 and 54,618 transcripts for the two species from leaf tissues, and conducted bioinformatics analyses to uncover 38 and 32 enzyme-encoding genes involved in phytoecdysteroids biosynthesis pathway. The analyses were done using standard methods and the conclusions are well-supported by the results. Moreover, the authors have made their data available, which will be very valuable for research community working on the species.
I have some comments that the authors could consider when revising the manuscript, as follows:
- Line 138: I think a more suitable word would be “remove” other than “delete”.
- Line 154: was BUSCO completeness increased or decreased from pre-processed transcripts to processed transcripts? Maybe it is worth checking this?
- How were the transcripts assembled for this study for the two species compared to other transcriptomes that were generated for other fern species available?
- Line 210: it would be more informative to provide with % of transcriptome were annotated for each species.
- Line 275: the authors discussed about redundant transcripts, for example, five AACT transcripts, and in the discussion (line 378) they speculated that these could be due to the whole genome duplication in fern species. This might be true, however, there is no experimental validation data supporting this yet in the current version. Could these be just “transcript isoforms” computationally predicted by Trinity? Did the authors used all transcript isoforms or only main transcripts from Trinity results?
- For qRT-PCR, to validate the RNA-Seq results, which RNA samples were used? Were they the same samples used for RNA-seq library preparation?
- For the case there were more than one transcript for one gene, the authors designed primers for the transcript with highest TPM. I assume that these transcripts are different isoforms of a same gene, and some parts were commonly shared. Were primers designed for the regions unique to the selected transcript, or for all of the transcripts that were annotated for the same gene?
- Line 279: the TPM values in text would be better with 2 decimal places.
- Would that be possible to identify genome-wide differentially expressed genes by comparing transcriptome data between the two species? Before moving on to genes in the PE pathways. This might point out pathways contributing to the different between the two species. What do the authors think about this?
- As the authors have noted in the discussion that, not only the number of genes, but also the expression levels could influence the traits. For transcription factors identified for each species, how many of them were differentially expressed between the two species?
- For visualization, I think it would be more informative to have the expression values from two species incorporated into Figure 8a, not just only the highest values. How about heatmaps next to each gene showing the (contrast) expression levels? I think panel 8b is just a validation of the RNA-seq data, and can be presented in the Supplementary.
Reviewer 2 Report
The paper is well written and the work is properly conducted. The authors report a metabolic profile for ecdysone and related compounds (PEs) in two different fern species Microsorum punctatum (MP) and M. scolopendria (MS). They show that the two species have 3 main PEs related to ecdysone in common but at different concentration and one PE specific of MS. Transcriptomic analysis and extensive bioinformatic characterization highlight differences between the two species in the number of identified enzymes of the putative PEs pathway. The authors also analyse CYPs genes and transcription factors that are relevant genes for biochemical pathways. The results will surely be useful for understanding the biosynthesis of PEs in the fern species and will also help synthesis of PEs in other systems for production purposes.
minor revisions:
lines 82-89: the authors report sterile fronds of MP and MS (which are then used for metabolite extraction) and fertile fronds only for MS, is MP always sterile?
line 99 analysis using a crude sample (2 mg dry weight) of Microsorum sterile fronds was used: was used to be removed, or re-phrase
line 100: The HPLC result of MP extract was performed peaks with the retention times of 3.283: was performed peaks is incorrect (showed peaks)
line 108: number to be replaced with concentration
lines 430-431: It may be that the up-regulation of genes encod- ing AACT1, HMGS1, IDI, ERG24, and DHCR24 in MS compared to MP might be related: may be and might, re-phrase
488 Calibration of the system with known amounts of these molecules allowed the concentration of these components in dry fronds was carried out on the basis of the stand- 489 ard samples (Sigma-Aldrich, St. Louis, MO, USA): the sentence should be re-phrased
Round 2
Reviewer 1 Report
I would like to thank the authors for adequately addressing my concerns from the previous round.
I have only 1 minor point that the authors could consider when revising their manuscript.
- Regarding the DEG identification part, I think it would be better if the author clearly explain in the respective section that why they used the MP transcriptome as the reference for their analysis to identify DEGs for the 2 species, MS and MP? While this is an acceptable way, a more suitable approach would probably be to pool the two transcriptome sets together, followed by redundancy deduction by clustering and use the resulting transcript set for DEG analysis.
